# Influence of Microalloying on the Microstructures and Properties of Spalling-Resistant Wheel Steel

**DOI:** 10.3390/ma16051972

**Published:** 2023-02-28

**Authors:** Tao Cong, Bo Jiang, Qiang Zou, Sancheng Yao

**Affiliations:** 1Metals & Chemistry Research Institute, China Academy of Railway Sciences Corporation Limited, Beijing 100081, China; 2Maanshan Iron & Steel Co., Ltd., Maanshan 243021, China

**Keywords:** rolling contact fatigue, wheel steel, ratcheting, microalloying, microstructure

## Abstract

Microalloyed steels have emerged to replace conventional plain-carbon steels to achieve longer wheel life on Chinese railroads. In this work, with the aim of preventing spalling, a mechanism that consists of ratcheting and shakedown theory correlated with steel properties is systematically investigated. Mechanical and ratcheting tests were carried out for microalloyed wheel steel to which vanadium was added in the range of 0–0.15 wt.% and the results were compared with that obtained for conventional plain-carbon wheel steel. The microstructure and precipitation were characterized via microscopy. As a result, the grain size was not obviously refined, and the pearlite lamellar spacing decreased from 148 nm to 131 nm in microalloyed wheel steel. Moreover, an increase in the number of vanadium carbide precipitates was observed, which were mainly dispersed and uneven, and precipitated in the pro-eutectoid ferrite region, in contrast to the observation of lower precipitation in the pearlite. It has been found that vanadium addition can lead to an increase in yield strength by precipitation strengthening, with no reduction or increase in tensile strength, elongation or hardness. The ratcheting strain rate for microalloyed wheel steel was determined to be lower than that for plain-carbon wheel steel via asymmetrical cyclic stressing tests. An increase in the pro-eutectoid ferrite content leads to beneficial wear, which can diminish spalling and surface-initiated RCF.

## 1. Introduction

Due to the greater axle load and higher traction of trains, especially for locomotives, the tread surfaces of wheels are aggravated by rolling contact fatigue (RCF) [1,2,3]. Cracks due to spalling are initiated from the tread surface and are generally caused by the accumulation of ratcheting strain [4]. Spalling is also caused by inclusions located at the subsurface of the wheel trade [5]. To prevent the development of cracks and spalling, the wheels are often replaced and reprofiled [6]. It is worth noting that the replacement and maintenance costs associated with RCF for revenue railroad wheels in China can be tens of millions of dollars, and the life of the wheel can also be reduced [7]. It has been found that the traction coefficient is increased with an increase in the slip ratio, and the position of maximum shear stress tends to occur near to the wheel surface simultaneously [8,9,10]. The locomotive conditions result in high traction forces being induced on a small contact patch on the field side of wheels and show a tendency for RCF damage [11]. Owing to the high traction coefficient, the occurrence of spalling is prevalent in locomotive wheels. The development of spalling-resistant wheel steel becomes more and more urgent.

Conventionally, as the carbon content of wheel steel is increased, the strength and hardness can be improved, but the concomitant decrease in toughness is unacceptable [12]. The advantage of using suitable microalloying steel has been studied by various experimental and numerical methods [13]. More specifically, microalloying steel shows a fine grain ferrite–pearlite microstructure with a grain size of 6–7 and can delay the initiation of fatigue cracks with a higher endurance limit [14]. After widespread investigation [15,16,17], microalloying steel is now regarded as one of the most competitive candidates in high-strength steels, due to its good balance with strength and toughness. The influence of microalloying in improving the spalling resistance of wheel steel, which is a type of low-carbon forged steel, has been rarely investigated.

Consequently, in this paper, based on the failure analysis of actual locomotive wheels, the mechanism for the RCF problem is systematically investigated, and an effective solution is proposed. Mechanical properties and ratcheting tests were carried out for plain-carbon and microalloying wheel steels with different strength levels. Likewise, the microstructure and properties of the two steels were compared using microscopy. The precipitates of microalloyed wheel steel with vanadium (V) were studied with the purpose of investigating the strength mechanism and contributions.

## 2. RCF Mechanism and Material Design

Fifteen wheels with spalling or tread cracks were removed from locomotives on a revenue railroad for detailed investigation. Examples of surface-initiated rolling contact fatigue for two wheels are shown in Figure 1. Two images show spalling, peeling and rolling contact fatigue cracks, which are regularly and continuously distributed circumferentially along the surface of the wheels.

With the aim of observing the location of initiation and plastic deformation, the crack surface was opened by cutting around the spalling, as shown in Figure 2a, and other cracks were radially cut from the wheel rims by a saw to prepare metallographic specimens, as shown in Figure 2b. Each specimen was mounted in resin, ground with 400–2000 grit emery papers, polished with diamond, etched with 4% nital and observed by optical microscopy (OM) (LEICA DMI4000). It is obvious from Figure 2a that cracks form on the tread surface and grow toward the inside with a definite angle. It is clear from Figure 2b that severe plastic deformation can occur on the surface layer of the wheel tread, and the formation process for the RCF cracks can be explained as follows: cracks form on the tread surface and propagate along the plastic deformation line. The tip of the crack remains in the deformation layer. However, the propagation directions for some crack tips change significantly with a tendency to expand to the interior of the wheels.

In view of the crack characteristics and the deformation of the microstructure of the wheel, all cracks start from the tread surface and are caused by plastic deformation. Thus, the essence of surface-initiated RCF cracks is the plastic deformation of materials. Due to this behavior, the shear yield strength and ratcheting effect of wheel materials should be discussed.

Under asymmetrical cyclic loads, the material has the following four forms [18]: perfect elasticity, elastic shakedown, plastic shakedown and ratcheting. When the load on the material exceeds the plastic shakedown limit, its plastic strain will continuously accumulate under cyclic loading. This process is called the ratcheting effect. Under the influence of the ratcheting effect, the toughness and fatigue properties of the material are constantly exhausted, which eventually leads to the initiation of RCF cracks.

The ratcheting effect describes the cyclic accumulation of the plastic deformation of materials or structures under asymmetric stress cyclic loading. The plastic deformation accumulation caused by the ratcheting effect is called ratcheting strain. In practice, there are many components or parts, such as the wheels of railroads, that will have a ratcheting effect under asymmetric cyclic loading. If the ratcheting effect occurs in the working state of a structure, excessive ratcheting strain will lead to the failure of the structure.

Wheel–rail contact can be simplified as a solid wheel rolling on an infinite plane. According to the shakedown map theory by Johnson et al. [19,20], the shakedown limit of the wheel can be obtained by Equation (1).
(1)$$\mu p_{0}/k_{e}<1$$

In addition, under the condition of pure rolling, there is a proportional relationship between the maximum shear stress and maximum contact pressure, where *p*_0_ = 4*k*_e_. Based on Formula (1), under a certain traction coefficient and load, the RCF performance is evidently determined by the yield strength of the wheel rim, which may be taken to be $\sqrt[{}]{3}$*k*_e_. Consequently, in terms of the material, improving the yield strength of wheels and decreasing the ratcheting strain are the best options to inhibit the initiation of rolling contact fatigue cracks.

Microalloying with V can enable one to refine the grain size and upgrade the plastic properties. V is usually used to modify material properties and microstructures by forming stable VC or VN precipitates [17]. Generally, the precipitation strengthening of steel can significantly benefit the strength of steel, but deteriorate its toughness [21,22]. For the case of microalloying steel, the strengthening effect of grain refinement is sufficient to eliminate the detrimental effect of precipitation. As a result, the grain refinement counteracts the decrease in toughness caused by precipitation strengthening. Thus, seeking an excellent balance is the merit of microalloying technology.

The development work conducted at Masteel Company on wheel steels has focused on improving the yield strength for a new wheel grade. This has been achieved by the addition of V (0.15% max) to the steel and optimizing the heat treatment, which is aimed at realizing adequate toughness. A higher yield strength can be achieved by keeping the carbon level at 0.67% max with suitable microalloying with V.

## 3. Experimental Procedures

### 3.1. Test Materials and Microstructure

The materials used in this investigation are based on two-wheel steels obtained from Chinese locomotives. The nominal chemical compositions of these steels are given in Table 1 and are labeled as J12 and J13 with the addition of 0.12% V. These two steels are similar in composition, except for their vanadium content. The wheels are forged and quenched with water spray to achieve an excellent rim property.

Small pieces with dimensions of 10 × 10 × 10 mm^3^ were cut from the two wheel rims. After carefully grinding and polishing, the samples were etched using 4% nital. The microstructures of the samples were examined using optical microscopy (OM) and scanning electron microscopy (SEM) (FEI-Quanta400). The optical micrographs obtained for J12 and J13 are shown in Figure 3. The microstructures for J12 and J13 consist of pearlite (gray color in Figure 3) and the pro-eutectoid ferrite (white color in Figure 3). Subsequently, the features of pearlite lamellae were observed for J12 and J13 using an FEI Quanta 200 FEG field-emission scanning electron microscope operated at a voltage of 20 kV. As shown in Figure 4, the SEM micrograph of the two wheel rims exhibits a fine pearlite microstructure at high magnification (3000×). The average volume fraction of the pro-eutectoid ferrite content was obtained from the OM images analyzed below, using a Clemex vision PE. Moreover, the pearlite lamellar spacing was measured using the SEM images and a linear truncation procedure. To ensure the accuracy of the results, at least fifty OM and SEM images were obtained for each sample cut from the two wheel rims.

To characterize the vanadium carbide (VC) precipitation and dispersion in detail, a transmission electron microscope (TEM, FEI Tecnai G2 F30) operated at 200 kV was employed. The TEM samples cut from the wheel rims at a depth of 15 mm from the tread surface were prepared inside the chamber of an FIB/SEM dual-beam system (FEI Helios NanoLab 600i).

Specimens for tensile tests were longitudinally cut from the wheel rim. Tensile tests were performed for the specimens with a gauge length and diameter of 10 mm and 50 mm at a strain rate of 10^−4^ s^−1^, using an MTS 810 testing machine at room temperature. For the reliability of the results, five specimens from the same location were tested to obtain the average value. The Brinell hardness was measured using a hardness test machine with a 10 mm test ball and a load of 3000 kgf. Fifteen measurements were performed at various locations on the wheel rim and the average value was determined. The mechanical properties of the wheel steels are presented in Table 2.

### 3.2. Uniaxial Ratcheting Test

In this section, a ratcheting effect experiment was performed on the wheel steels J12 and J13, and an asymmetrical cyclic stressing test was performed using a single-shaft fatigue testing machine. The evolution curve of the ratcheting behavior was obtained from the experimental results, and the evolution pattern of the ratcheting behavior was compared for the two kinds of wheel steels.

The cyclic experimental method of stress control was used in the experiments. The equipment consisted of a single-shaft material testing machine MTS 810, which can be used to automatically control and collect real-time experimental data. The whole process can be considered to involve quasi-static tension. The loading rate for the stress cycle experiment was 100 MPa/s. The change in axial strain was monitored using a strain meter. The sampling position for the materials in the experiment is shown in Figure 5. Five cylindrical bar-shaped specimens were taken at a distance of 15 mm away from the nominal rolling circle of the tread. The specimens were then processed into threaded fatigue specimens. The detailed processing dimensions are shown in Figure 6, in which the parallel segment has a diameter of 10 mm and the clamping end has an M16 tread.

Judging the yield strength and tensile strength for wheel steels J12 and J13 (as shown in Table 2), the stress level of the ratcheting effect experiment should be higher than the yield strength and lower than the tensile strength. Therefore, the stress level in this experiment should range between 400 and 1100 MPa. Since the load of the ratcheting effect is asymmetric, the mean stress needs to be determined in advance. The average stress in this experiment was determined to be 100 MPa. During the experiment, the stress varied over a range of 400–800 MPa, depending on the steel condition. All experiments ended with the specimen being pulled apart. The fatigue life of the specimen was quantified based on the number of cycles completed before the failure of the specimen. The maximum number of cycles for a single fatigue test was set to 10,100. When the number of cycles reaches 10,100, the steel can be considered to reach a plastic stable state under the current average stress and stress amplitude.

For a general uniaxial stress cycle experiment, the ratcheting strain produced in the experiment can be defined by multiple methods. In this paper, the ratcheting strain is defined by the maximum axial strain in each cycle that corresponds to each stress–strain hysteresis loop, as shown in Equation (2). To further discuss the evolution pattern for ratcheting behavior with the number of cycles, a ratcheting strain rate is proposed. The ratcheting strain rate $d\varepsilon_{r}/dN$ is the increment in the ratcheting strain in each cycle, where N represents the number of cycles, as shown in Equation (3) [23].
(2)$$\varepsilon_{r}=\varepsilon_{\max}$$
(3)$$\overset{}{{\overset{•}{\varepsilon }}_{r}}=\frac{d\varepsilon_{r}}{dN}$$

## 4. Results

### 4.1. Mechanical Properties and Unixial Ratcheting

The mechanical properties obtained for the specimens are given in Table 2. The yield strength of J13 steel is 13% greater than that of J12 steel. It is evident that the values of tensile strength, elongation and hardness are in the same range for both wheels.

Figure 7 shows the ratcheting evolution curves for the J12 and J13 wheel steels. The rate of ratcheting strain on the steel increases with the number of cycles. Under a general stress load, the ratcheting strain increases almost linearly with the number of cycles. However, when the number of cycles approaches the fatigue life of the specimen, the ratcheting strain rate suddenly increases and the slope of the curve increases sharply, followed by failure over a short time. The ratcheting evolution curves obtained for the two kinds of wheel steels are consistent with those obtained for 25CDV4.11 steel under different stress levels, and can be divided into three stages [23].

The slope of the ratcheting evolution curve often reflects the ratcheting resistance of different steels. The smaller the slope, the smaller the ratcheting strain rate and the better the ratcheting behavior of the steel at the same stress amplitude. It can be judged that steel has a better spalling resistance in practice. Figure 7 shows that the slope of the evolution curve for J12 ratcheting is larger than that for J13 ratcheting under different stress amplitudes of fixed average stress; that is, the ratcheting strain rate for J13 is smaller.

By analyzing the experimental results obtained for the ratcheting effect for two kinds of wheel steels (as shown in Table 3), it is found that J12 and J13 steels enter the plastic stable state at 100 ± 400~600 MPa and 100 ± 500~600 MPa near the stability limit, respectively. At the same time, considering the fatigue life of the two steels under different stress amplitudes with fixed average stress, it can also be concluded that the ratcheting strain rate for J13 is lower than that for J12, indicating better ratcheting resistance performance. Figure 8 and Figure 9 show the fracture surfaces obtained following the ratcheting tests for J12 and J13, respectively. A typical fracture surface morphology for low-cycle fatigue is observed, with many dimples formed by cyclic plastic deformation.

In summary, the ratcheting rate for J13 is lower in correspondence with the yield strength, and is appreciably higher than that for J12.

### 4.2. Materials and Microstructure Characteristics

Since the volume fraction of the pro-eutectoid ferrite, the grain size and the spacing of the lamellar for steel play an important role in the mechanical properties of steel and ratcheting, it is necessary to compare these properties for both J12 and J13 steels.

OM and SEM reveal a microstructure composed of the pro-eutectoid ferrite and pearlite in both wheel steels (Figure 3 and Figure 4). The measured microstructural parameters based on OM and SEM micrographs, including the volume fraction of the pro-eutectoid ferrite, grain size and spacing of the lamellar, are given in Table 4. The average volume fractions of the pro-eutectoid ferrite in J12 and J13 wheel steel are 2.3% and 5.8%, respectively. The grain size for J12 and J13 was measured using 50 samples, and the mean values (33 μm and 29 μm) are listed. The spacing of the J12 lamella is measured to be 148 nm on average, and the average spacing in J13 is reduced accordingly to 131 nm (a decrease of 17 nm). This phenomenon clearly indicates that the grain size is not reduced by the addition of V, but a thinner spacing of the lamellar and a higher amount of pro-eutectoid ferrite, in contrast to J12 steel [24].

Figure 10a–c show that the precipitates are absent in J12 steel without V addition and that the cementite laths are continuous. TEM micrographs of J13 steel reveal the fine dispersion of precipitates (as marked by yellow cycles in Figure 11a–c), which was identified as VC in the pro-eutectoid ferrite regions and in pearlitic ferrite lamellae. The J13 precipitates are randomly distributed in the pro-eutectoid ferrite and do not appear with a straight or certain orientation in relationship to the ferrite matrix [25].

It is also observed that microalloying with V for the J13 steel leads to precipitates in the pro-eutectoid ferrite that form a weaker phase with some precipitates in the pearlitic ferrite, as illustrated in Figure 11a–c, which is the principal strengthening mechanism for wheel yield strength. V can significantly improve the yield strength by directly obstructing and more effectively pinning the movement of dislocations, which is attributed to the interaction between fine and dispersed precipitates and dislocations in ferrite. The precipitation occurs primarily at the austenite/ferrite transformation interface, which is termed interphase precipitation and displays characteristics that are similar to other lower-carbon microalloying steels, as reported previously [26].

### 4.3. The Influence of VC Precipitates on the Ratcheting Property of Wheel Steel

The results show the presence of VC precipitates in both the pro-eutectoid ferrite regions and the pearlitic ferrite lamellae in J13 steel that contain 0.12% V, consistent with precipitation strengthening. In terms of precipitation strengthening mainly in the pro-eutectoid ferrite, it is noted that the yield strength of J13 steel, rather than its tensile strength and hardness, shows a marginal increase, which is beneficial to RCF resistance. In addition, the yield strength of J13 increased significantly more compared to its tensile strength. V addition can enhance the yield strength, while the tensile strength and hardness can remain at the same level, as is the case for plain carbon (J12) with an optimized chemical composition. This enhancement is achieved without any reduction in its toughness.

Furthermore, it is interesting to find that due to the effect of V, the width of the cementite laths is reduced, and the cementite boundary becomes discontinuous in the pearlite phase in J13, which leads to a significant increase in the amount of the pro-eutectoid ferrite and ferrite in the pearlite phase in J13, with respect to that in J12 (Table 4). Because V addition contributes to the enlargement of the ferrite phase area, during the transformation of undercooled austenite, the driving force for ferrite transformation increases, eventually followed by the formation of 152% more pro-eutectoid ferrite in J13 than in J12 and a reduction in the amount of cementite.

In generally, the precipitation strengthening of steel can significantly enhance the strength of steel; however, it can also reduce the toughness of steel. Even though microalloying wheel steel with coarse V carbides can adversely influence its toughness, in this work, it can partly be compensated for by suitable heat treatment of J13 through refinement of the pearlite lamella.

Ferrite is regarded as a soft microstructure that decreases the wear resistance. Nevertheless, it is well known that a certain amount of wear is not harmful. In recent years, a competitive concept for wear and RCF has been proposed [27,28]. As previously discussed, microalloying steel with V can evidently increase the yield strength, tending to limit the initiation of RCF cracks formed by wheel/rail contact stress by decreasing the ratcheting rate. With respect to plain-carbon wheel steel, a higher yield strength is achieved with an inevitable increase in tensile strength and hardness. However, despite the presence of a thinner plastic deformation layer beneath the tread surface, formed by achieving a higher yield strength level, if the tensile strength and hardness increase, resulting in a reduction in the wear rate, the removal rate for RCF cracks cannot catch up with the propagation rate. The surface layer of material that contains RCF cracks cannot be completely or partially removed. Thus, there is no doubt that microalloyed wheel steel offers an available and effective strengthening method. Moreover, to ensure the improvement effect for J13, it is also necessary to balance the competitive relationship between RCF and wear; moreover, more ferrite tends to increase the wear rate, consequently slowing crack propagation. It is demonstrated that the RCF cracks on the J13 surface, rather than that for J12, can be easily removed and can have a significant effect on reducing the hazard of spalling.

With respect to the microstructure of wheel steel, not only can one consider strengthening the pro-eutectoid ferrite to inhibit crack initiation, but an increase in the ferrite content can lead to a reasonable increase in the wear rate to alleviate crack propagation at a certain depth or prevent it from branching toward the tread surface.

The microstructure and properties of J13 are modified to overcome ratcheting and crack growth that can lead to spalling or serious damage. In reality, however, the presence of water and high humidity results in a fluid-pumping effect. This accelerating factor for crack growth is non-negligible and is considered as sufficient to promote crack spalling in a revenue railroad. Further investigation of the practical effectiveness involved in the performance of microalloying wheels still needs to be carried out to gain future knowledge.

Therefore, in this paper, based on the failure analysis of actual locomotive wheels, the mechanism for the RCF problem is systematically investigated, and a beneficial solution is proposed. Mechanical properties and ratcheting tests were carried out for plain-carbon and microalloyed wheel steels with different strength levels. Likewise, the microstructure and properties of the two steels were compared using microscopy. The precipitates of microalloyed wheel steel with vanadium (V) were studied with the purpose of investigating the strength mechanism and contributions.

## 5. Conclusions

The influence of microalloying on the microstructures and properties of spalling-resistant wheel steel was investigated. The results can be summarized as follows:(1)The spalling of the locomotive wheel treads is attributed to the ratcheting mechanism. To prevent ratcheting and the formation of spalling, microalloying was applied to design a wheel steel.(2)The microalloyed wheel steel J13 shows superior yield strength with no reduction in toughness, compared with that for the plain-carbon steel J12. The yield strength of J13 is directly influenced by precipitation strengthening in both pro-eutectoid and pearlitic ferrite.(3)The wheel steel J13 shows a lower ratcheting strain rate under the same stress, consistent with its higher yield strength, which can mitigate the accumulation of plastic deformation that leads to ratcheting effects.(4)The volume fraction of pro-eutectoid ferrite in J13 is the microstructural mechanism that improves the mechanical properties of the steel, including the ratcheting effect.

## Figures and Tables

**Figure 1 materials-16-01972-f001:**
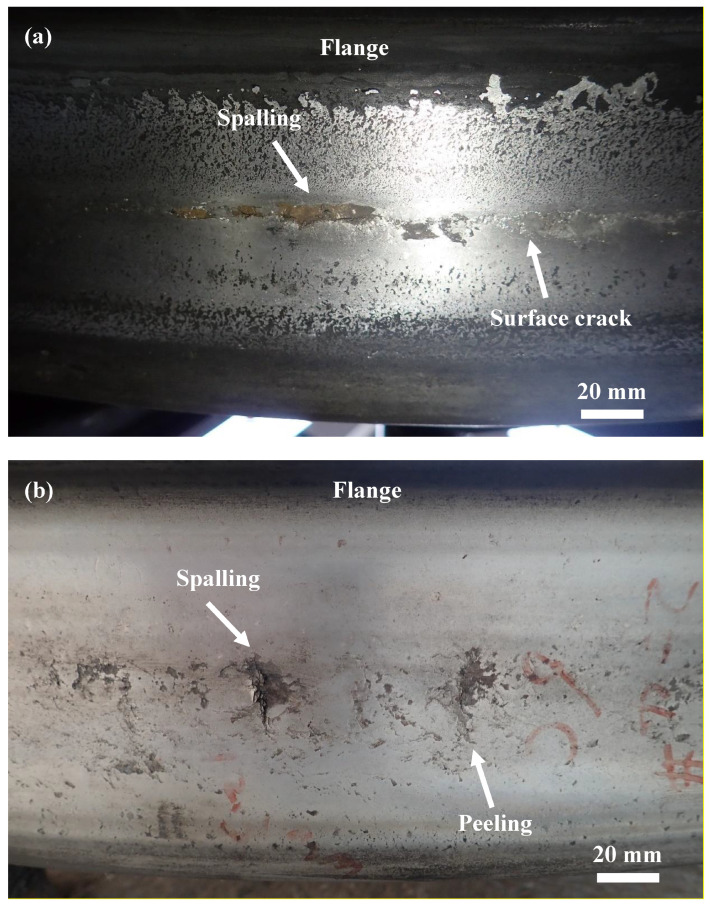
Two typical surface-initiated rolling contact fatigue features for wheels that show (**a**) spalling and cracking and (**b**) spalling, peeling and cracking.

**Figure 2 materials-16-01972-f002:**
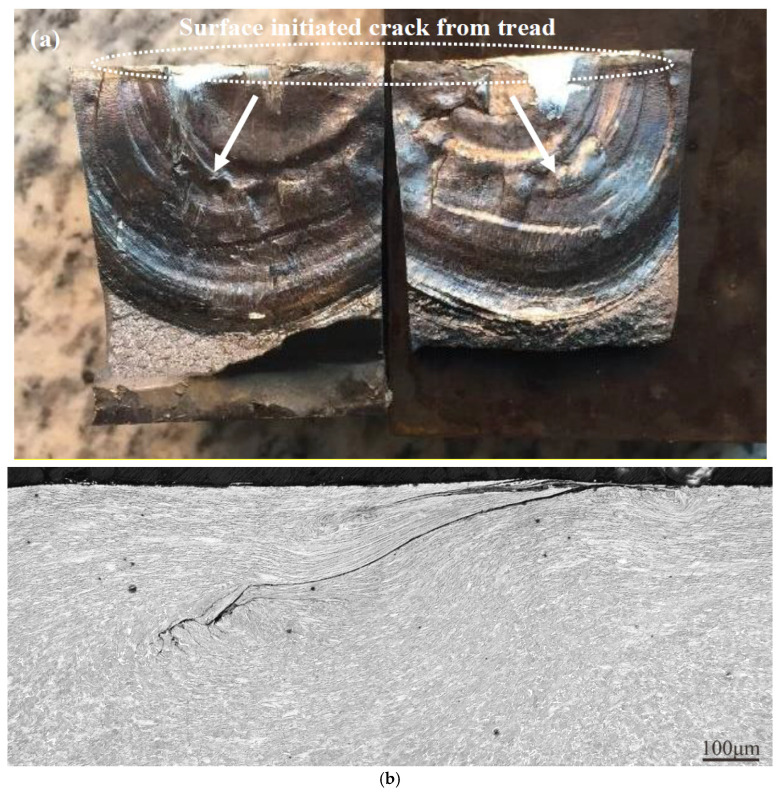
Typical rolling contact fatigue in wheels: (**a**) opening crack surface; (**b**) crack path and plastic deformation.

**Figure 3 materials-16-01972-f003:**
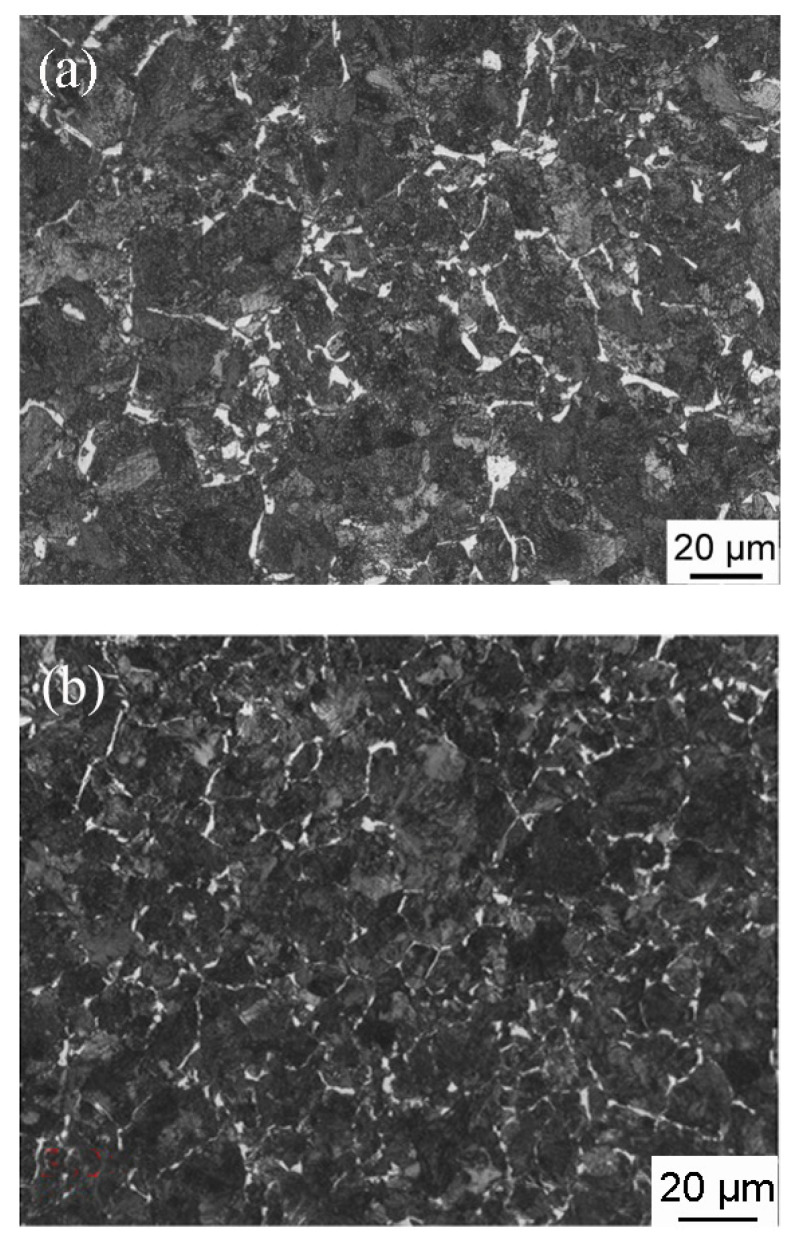
OM micrographs of the two tested wheel steels, (**a**) J12 and (**b**) J13.

**Figure 4 materials-16-01972-f004:**
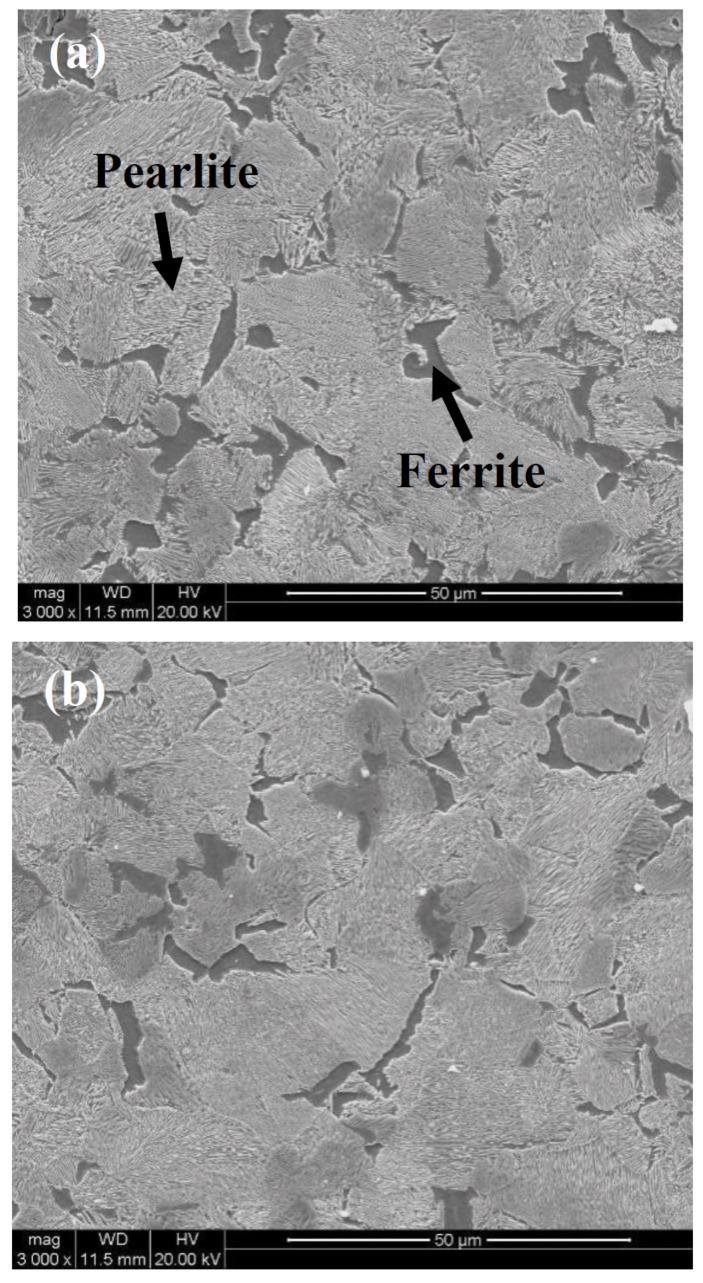
SEM micrographs for the two tested wheel steels, (**a**) J12 and (**b**) J13.

**Figure 5 materials-16-01972-f005:**
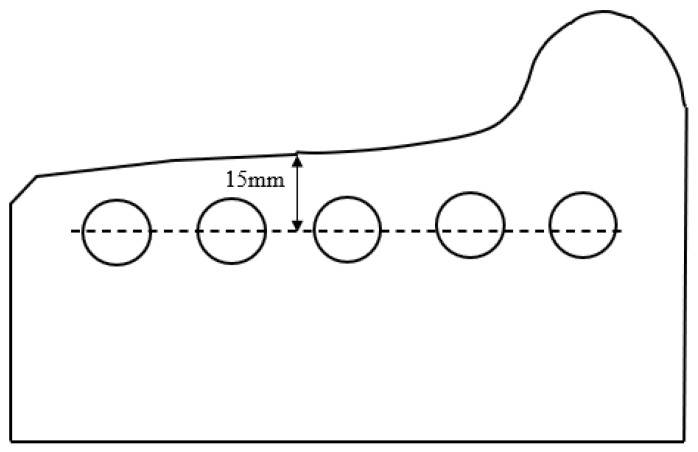
Schematic diagram of the specimen sampling location.

**Figure 6 materials-16-01972-f006:**
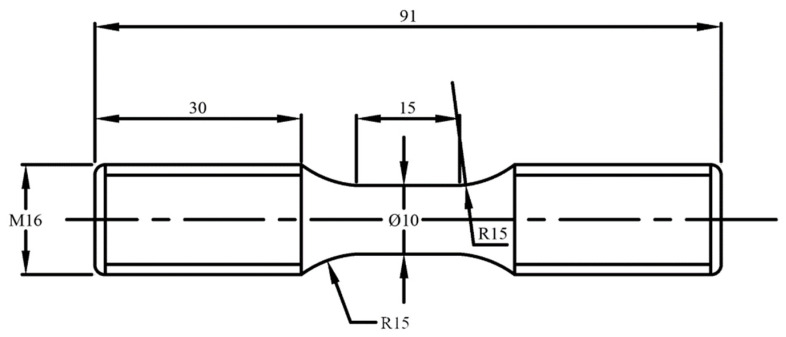
Dimensions of the ratcheting test specimens (units: mm).

**Figure 7 materials-16-01972-f007:**
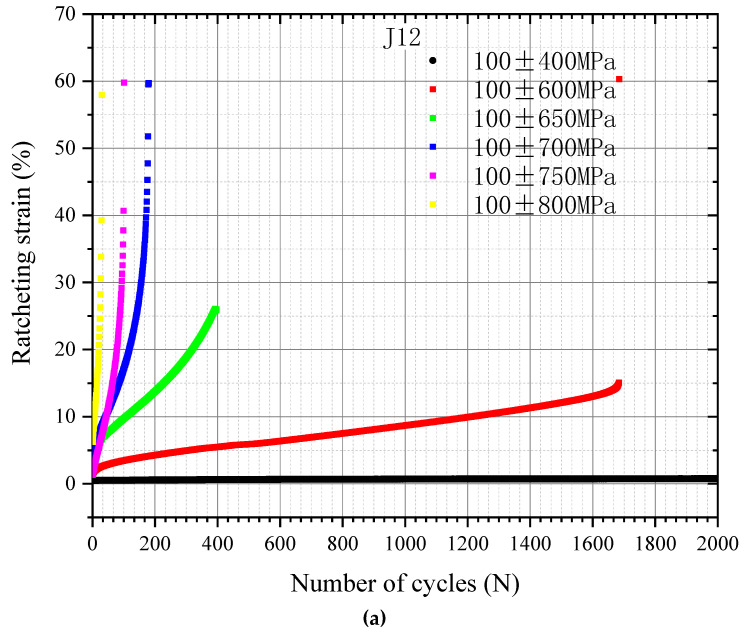
Ratcheting evolution curve for the following wheel steels: (**a**) J12; (**b**) J13.

**Figure 8 materials-16-01972-f008:**
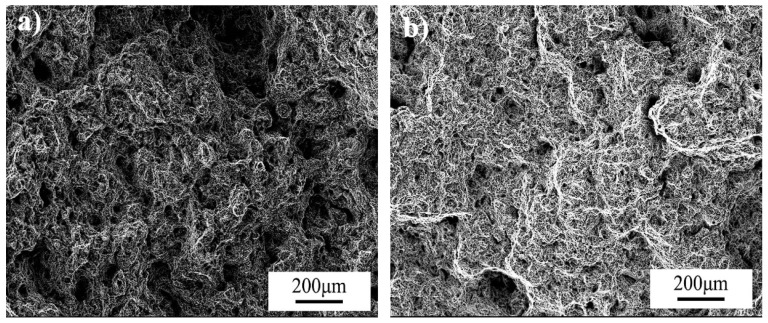
Fracture surface following ratcheting tests for J12: (**a**) 100 ± 800 MPa; (**b**) 100 ± 700 MPa.

**Figure 9 materials-16-01972-f009:**
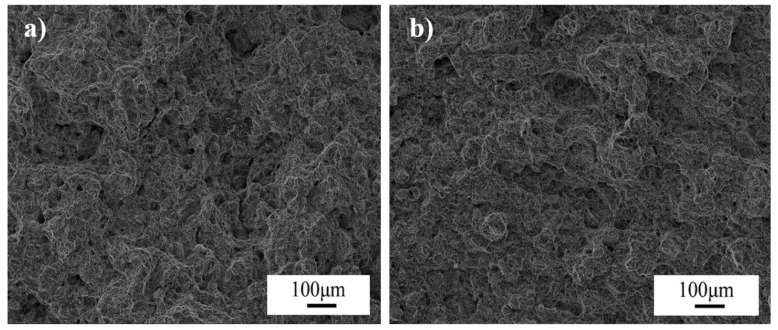
Fracture surface following ratcheting tests for J12: (**a**) 100 ± 800 MPa; (**b**) 100 ± 750 MPa.

**Figure 10 materials-16-01972-f010:**
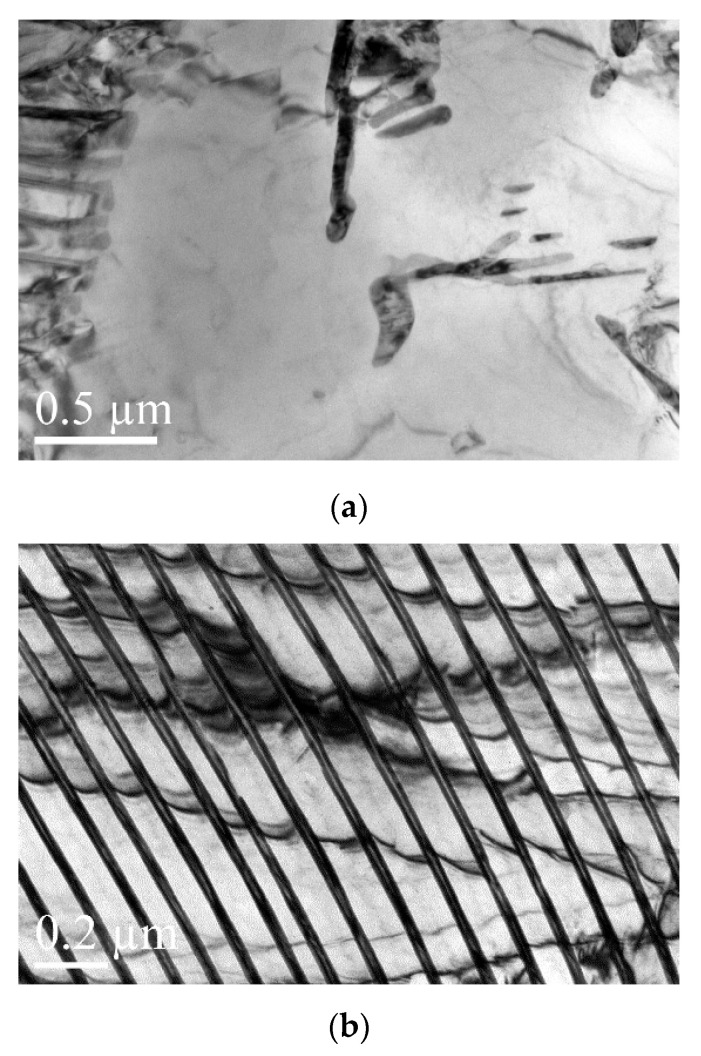
TEM micrographs for J12 that show (**a**) the pro-eutectoid ferrite and pearlite and (**b**,**c**) pearlite.

**Figure 11 materials-16-01972-f011:**
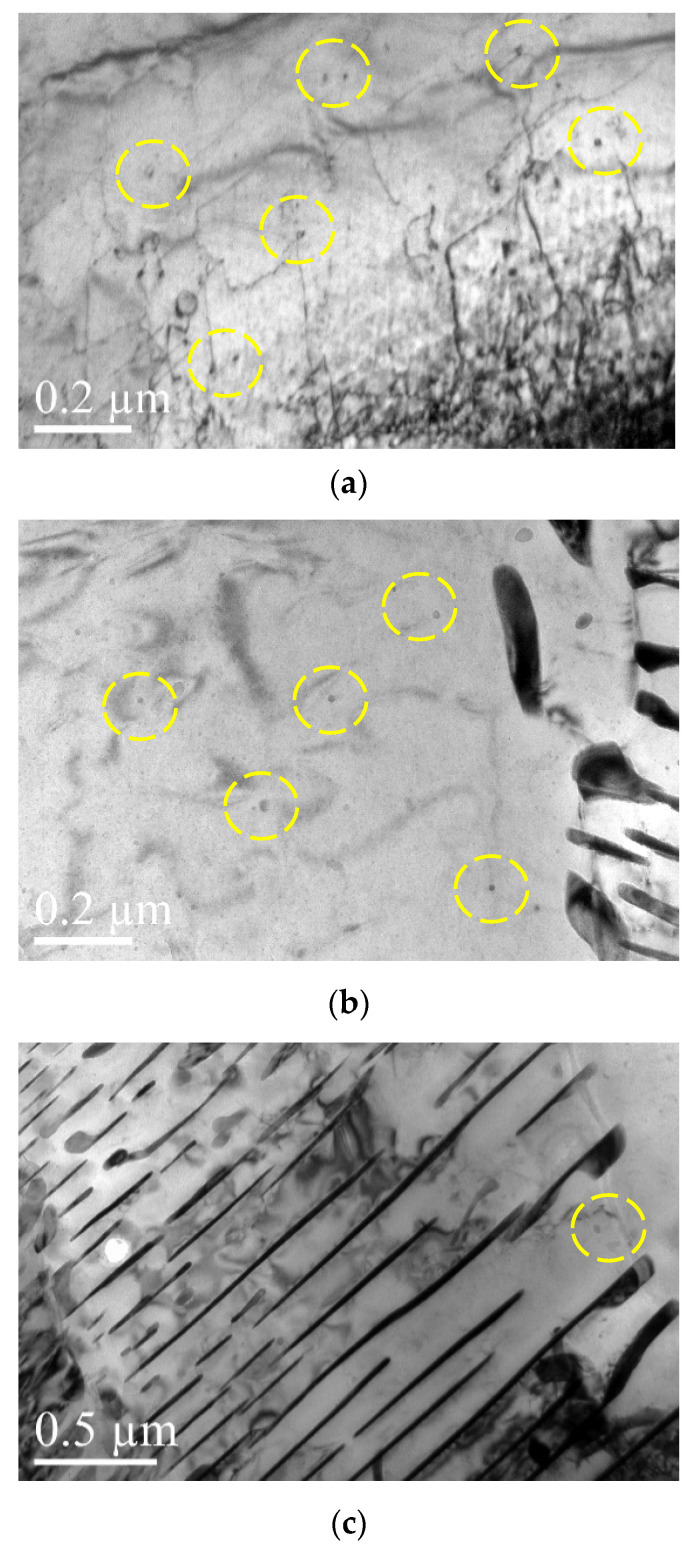
TEM micrographs for J13 that show the precipitates (**a**,**b**) in the pro-eutectoid ferrite and (**c**) in the pearlite.

**Table 1 materials-16-01972-t001:** The chemical compositions (mass percentage, wt.%).

	C	Si	Mn	P	S	V
J12	0.64	0.29	0.75	0.009	0.013	\
J13	0.64	0.25	0.73	0.008	0.012	0.12

**Table 2 materials-16-01972-t002:** Mechanical properties of the wheel steels.

	Yield Strength (MPa)	Tensile Strength (MPa)	Elongation %	Hardness (HB)
J12	691	1079	15	315
J13	781	1076	14	318

**Table 3 materials-16-01972-t003:** Relationship between stress amplitude and cycle order obtained from a ratcheting test for the two different wheel steels.

Stress Level	100 ± 800 MPa	100 ± 750 MPa	100 ± 700 MPa	100 ± 650 MPa	100 ± 600 MPa	100 ± 500 MPa	100 ± 400 MPa
J12	29	100	178	397	1684	-	>10,100
J13	174	291	410	1613	3396	>10,100	-

**Table 4 materials-16-01972-t004:** The difference in microstructural parameters for the two types of wheel steel.

	Volume Fraction of the Pro-Eutectoid Ferrite	Grain Size (μm)	Spacing of the Lamellar (nm)
J12	2.3	33	148
J13	5.8	29	131

## Data Availability

Data is unavailable due to privacy or ethical restrictions.

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
