# Peer review of "Influence of Microalloying on the Microstructures and Properties of Spalling-Resistant Wheel Steel"

_materials, 2023, doi:10.3390/ma16051972_

Round 1
Reviewer 1 Report
The article highlights peculiarities of mechanical behavior of microalloyed wheel steel to which vanadium was added in the range of 0-0.15 wt%, and the results are compared with that obtained for conventional plain-carbon wheel steel. The authors used the modern equipment for visualization and assistance in the interpretation of the obtained results. They showed that the microalloyed wheel steel J13 shows superior yield strength with no reduction in toughness compared with that for plain-carbon steel J12.
The article is interesting, but a number of shortcomings need to be corrected:
1. The authors note that “Cracks due to spalling are initiated from the tread surface…”, however, the authors should note in the text of the article that cracks can initiate not only from the wheel surface, but also from the wheel subsurface (for example, the following reference can be used: https://doi.org/10.1007/s11003-013-9557-7). Fatigue cracks may appear both on the rolling surfaces and in the subsurface layers. They grow, in turn, at different combinations of the processes of normal tension (which controls the range ΔKI ) and transverse shear (ΔKII).
2. The authors should add references to confirm the reasoning given in the article “Generally, precipitation strengthening of steel can significantly benefit the strength of steel but deteriorate the toughness of steel”.
3. In Fig. 3, the scale bar cannot be recognized. This should be corrected.
4. The authors should correct the chapter numbering as there are two Chapters No. 3 (page 9 and page 15).
5. The authors should explain why the research was conducted under low-cycle fatigue conditions, since railway wheels work under high-cycle fatigue conditions.
6. The authors should explain why they received a fracture surface with many dimples formed. This class of wheel steels is characterized by the presence of intragranular cleavage facets, which are not observed in this case.
7. It is desirable for the authors to add references for confirming obtained results of reducing the distance between the lamellae for vanadium microalloyed steel “The spacing of the J12 lamella is measured to be 148 nm on average, and the average spacing in J13 is reduced accordingly to 131 nm (a decrease of 17 nm).”
Author Response
- The authors note that “Cracks due to spalling are initiated from the tread surface…”, however, the authors should note in the text of the article that cracks can initiate not only from the wheel surface, but also from the wheel subsurface (for example, the following reference can be used: https://doi.org/10.1007/s11003-013-9557-7). Fatigue cracks may appear both on the rolling surfaces and in the subsurface layers. They grow, in turn, at different combinations of the processes of normal tension (which controls the range ΔKI ) and transverse shear (ΔKII).
Response: Thanks for the comments. We have revised the paper according to the suggestions. (highlighted by color)
- The authors should add references to confirm the reasoning given in the article “Generally, precipitation strengthening of steel can significantly benefit the strength of steel but deteriorate the toughness of steel”.
Response: According to the reviewer’s comments, references has been added. (highlighted by color)
- In Fig. 3, the scale bar cannot be recognized. This should be corrected.
Response: Thanks for the comments. We have revised it. (highlighted by color)
- The authors should correct the chapter numbering as there are two Chapters No. 3 (page 9 and page 15).
Response: According to the useful comments, we have revised the paper. (highlighted by color)
- The authors should explain why the research was conducted under low-cycle fatigue conditions, since railway wheels work under high-cycle fatigue conditions.
Response: We thank the reviewer for the constructive comments. In section 2, the failure mechanism of spalling is caused by ratcheting effect. Under asymmetrical cyclic loads, the material has four forms: perfect elasticity, elastic shakedown, plastic shakedown and ratcheting. When the load on the material exceeds the plastic shakedown limit, its plastic strain will be continuously accumulated under cyclic loading. This process is called the ratcheting effect. Our tests in this paper is the ratcheting tests to obtain the shakedown limit of the wheel steel. (highlighted by color)
- The authors should explain why they received a fracture surface with many dimples formed. This class of wheel steels is characterized by the presence of intragranular cleavage facets, which are not observed in this case.
Response: Thanks for the comments. The applied stress are 100±800 MPa and 100±700 MPa which are larger than the yield strength of the wheel steel (691 MPa of J12, 781 MPa of J13). (highlighted by color)
- It is desirable for the authors to add references for confirming obtained results of reducing the distance between the lamellae for vanadium microalloyed steel “The spacing of the J12 lamella is measured to be 148 nm on average, and the average spacing in J13 is reduced accordingly to 131 nm (a decrease of 17 nm).”
Response: According to the reviewer’s comments, references has been added. (highlighted by color)

Reviewer 2 Report
The submitted study is reviewed carefully.
1) The reviewer is very disappointed by al the weaknesses of this paper and strongly suggests to the authors to greatly improve the scientific content of a possible next paper on this topic, both on the definition of the model, and the numerical / mechanical content of the results.
Also, the following comments are needed to be addressed to improve this paper:
2) In the introduction section, not only to tell the readers who has done what, the authors also should tell the readers what the differences are between your work and the published literatures. .
3) The theoretical contributes and tools described in the paper are not new. The mathematical model was not developed by the author; it contains equations from the literature. Nevertheless, the author does not mention the source of the equations presented.
4) The author should spend more time preparing the graphs presented. None of the diagrams have sufficient labeling
5) Some conclusions cannot give some results and explanation. So, the conclusions should be rewritten and simplified.
6) References are not up-to-date and adequate in the original manuscript, you should revise this part.
7) There are still grammar errors and typos in the text and they should be corrected by an author who is native English or has better writing skills.
Author Response
1) The reviewer is very disappointed by al the weaknesses of this paper and strongly suggests to the authors to greatly improve the scientific content of a possible next paper on this topic, both on the definition of the model, and the numerical / mechanical content of the results.
Response: We thank the reviewer for the constructive comments. We will improve the scientific content including the model, and the numerical / mechanical content of the results in our next paper on this topic in future. (highlighted by color)
2) In the introduction section, not only to tell the readers who has done what, the authors also should tell the readers what the differences are between your work and the published literatures.
Response: According to the reviewer’s comments, we have added the differences between our work and the published literatures. (highlighted by color)
3) The theoretical contributes and tools described in the paper are not new. The mathematical model was not developed by the author; it contains equations from the literature. Nevertheless, the author does not mention the source of the equations presented.
Response: We thank the reviewer for the constructive comments. We have rewritten this part and added the reference. (highlighted by color)
4) The author should spend more time preparing the graphs presented. None of the diagrams have sufficient labeling
Response: Thanks for the comments. We have revised the paper according to the suggestions. (highlighted by color)
5) Some conclusions cannot give some results and explanation. So, the conclusions should be rewritten and simplified.
Response: Thanks for the comments. We have rewritten the conclusion. (highlighted by color)
6) References are not up-to-date and adequate in the original manuscript, you should revise this part.
Response: According to the useful comments, we have revised the paper. (highlighted by color)
7) There are still grammar errors and typos in the text and they should be corrected by an author who is native English or has better writing skills.
Response: Thanks for the comments. We have improved the language. (highlighted by color)

Round 2
Reviewer 1 Report
The authors took into account all comments of the reviewer and made appropriate corrections to the manuscript.
Reviewer 2 Report
Accept